# Tolerance and Recovery of Ultralow-Loaded Platinum Anode Electrodes upon Carbon Monoxide and Hydrogen Sulfide Exposure

**DOI:** 10.3390/molecules24193514

**Published:** 2019-09-27

**Authors:** Sebastian Prass, Kaspar Andreas Friedrich, Nada Zamel

**Affiliations:** 1Fraunhofer Institute for Solar Energy Systems ISE, Heidenhofstr. 2, 79110 Freiburg, Germany; nada.zamel@ise.fraunhofer.de; 2German Aerospace Center, Institute of Engineering Thermodynamics, Pfaffenwaldring 38-40, 70569 Stuttgart, Germany; andreas.friedrich@dlr.de; 3University of Stuttgart, Institute of Building Energetics, Thermal Engineering and Energy Storage (IGTE), Pfaffenwaldring 31, 70569 Stuttgart, Germany

**Keywords:** fuel impurities, ISO concentration, ultralow-loaded anode catalyst layer, platinum electrode, shut-down and start-up process

## Abstract

The effects of carbon monoxide (CO) and hydrogen sulfide (H_2_S) in concentrations close to their respective limits in the Hydrogen Quality Standard ISO 14687-2:2012 on the performance of proton exchange membrane fuel cells (PEMFCs) with ultralow-loaded platinum anode catalyst layers (CLs) were investigated. The anodic loadings were 50, 25, and 15 µg/cm^2^, which represent the current state-of-the-art, target, and stretch target, respectively, for future automotive PEMFCs. Additionally, the effect of shut-down and start-up (SD/SU) processes on recovery from sulfur poisoning was investigated. CO at an ISO concentration of 0.2 ppm caused severe voltage losses of ~40–50% for ultralow-loaded anode CLs. When H_2_S was in the fuel, these anode CLs exhibited both a nonlinear decrease in tolerance toward sulfur and an improved self-recovery during shut-down and start-up (SD/SU) processes. This observation was hypothesized to have resulted from the decrease in the ratio between CL thickness and geometric cell area, as interfacial effects of water in the pores increasingly impacted the performance of ultrathin CLs. The results indicate that during the next discussions on the Hydrogen Quality Standard, a reduction in the CO limit could be a reasonable alternative considering future PEMFC anodic loadings, while the H_2_S limit might not require modification.

## 1. Introduction

Proton exchange membrane fuel cells (PEMFCs) are a promising clean energy alternative for applications in the transport sector, as they combine high-power density and efficiency with the significant advantage of fast system refueling times. Hydrogen (H_2_) as a fuel might, however, contain low concentrations of impurities stemming from production and infrastructure. Impurities such as carbon monoxide (CO) and hydrogen sulfide (H_2_S) can deteriorate the performance and lifetime of PEMFCs. Naturally, the severity of an impurity is not only affected by its concentration (or rather, dose), but also by the catalyst type, operational parameters, cross-effects, and active or passive mitigation strategies [1,2]. For example, air-bleeding is an effective strategy to provide oxygen (O_2_) for the oxidation of adsorbed contaminant species in the anode electrode [3], while catalyst alloys containing platinum (Pt) and other platinum group metals (PGMs) can provide higher tolerances versus certain contaminants [4,5,6,7]. Although they are very effective, such mitigation strategies partially come with implications about performance or durability. For example, a fraction of the O_2_ introduced by air-bleeding readily reacts with H_2_ in the anode compartment and thereby lowers the fuel efficiency while simultaneously accelerating membrane degradation through additional peroxide and radical formation [8]. Moreover, alloy catalysts containing PGMs or metals other than Pt usually offer a lower stability, as the alloying components exhibit higher leaching rates. What typically remains is a catalyst particle with a Pt-enriched surface [9], while the leaching cations eventually have impacts on the protonic conductivity or even integrity of the ionomer in the electrode or membrane [10].

Apart from active or passive PEMFC system internal contamination mitigation techniques, adjusting the allowed impurity limits in the Hydrogen Quality Standard ISO 14687-2:2012 poses an additional layer in accommodating enhanced PEMFC requirements versus fuel contaminants. If electrode design or system internal strategies are exhausted, the allowed impurity level for the respective contaminant could be lowered at reasonable levels based upon tangible experimental PEMFC data. Although this option eventually leads to higher H_2_ production costs, it helps to avoid higher PEMFC system costs per vehicle or implications coming from internal tolerance improvement strategies.

Some of the major cost drivers in mass-produced PEMFC vehicles are the catalyst layers (CLs) attached to the membrane. The choice of CL materials, the electrode design, and production are primary levers in reducing PEMFC costs while simultaneously increasing the lifetime. Although substantial reductions in PGM catalyst loading per cell area have already been achieved, further reductions are required as a consequence of increasing PGM prizes with higher FC vehicle market penetration. The stipulated reductions range from 50% to 75% compared to the approximate state-of-the-art, resulting in PGM targets for 2020 of about 125 and 62.5 µg/cm^2^ depending on the contemplated scenario [11]. In both cases, the loading of the anode electrode is expected to account for 20% (i.e., 25 and 12.5 µg/cm^2^ of PGMs, respectively): this is called ultralow loading in the present study hereafter.

Generally, lower anodic catalyst loadings are less tolerant toward catalyst contaminants, as both fuel and contaminants compete for fewer active sites in the electrode. For pure Pt electrodes, the voltage drop was found to increase by 25% when the Pt-loading decreased from 400 to 50 µg/cm^2^ if 1 ppm CO was introduced [12,13]. A similar trend was observed for H_2_S, where the tolerance of the electrode was found to decreased proportionally with the reduction in the anode loading [14]. It is expected that this trend would continue for ultralow loadings (<50 µg_PGM_/cm^2^), but so far there has been no study in the literature that has investigated the tolerance of such ultralow anodic loadings. Additionally, processes such as the shut-down and start-up (SD/SU) of FC vehicles are expected to affect the poisoning phenomenon of the electrodes. During downtime, reactants can diffuse from the anode to the cathode, and conversely, mixed potentials arise at the electrodes and poisoned catalysts eventually recover. However, there are limited experimental data available in the literature on recovery due to SD/SU processes, which is especially of interest in the case of recovery from sulfur contamination. Cyclic voltammetry (CV)-like methods triggering oxidative processes at ~0.9–1.1 V count as a recovery strategy for sulfur-contaminated electrodes [6,15,16], but this strategy also induces carbon corrosion and therefore destruction of the electrode itself.

The study presented here therefore seeks to add to the studies by Hashimasa et al. [12,14] by investigating the tolerance of ultralow-loaded anodic platinum catalyst layers. Two different types of contaminants were selected: CO, as its poisoning effect is fully reversible, while in contrast, H_2_S typically poisons the catalyst irreversibly during regular fuel cell operation. Additionally, recovery from sulfur poisoning through simple shut-down and start-up (SD/SU) processes was examined in more detail for ultralow anodic catalyst loadings.

## 2. Materials and Methods

### 2.1. Test Station and Contaminant Introduction

Single-cell tests were carried out in an in-house-built test station with an integrated potentiostat (Zahner Zennium Pro) and an electric load (Kikusui PLZ664WA) with fluidics (shown schematically in Figure 1).

In principle, the test station was comparable to the one used by Hashimasa et al. [12], but with a different humidification system for the anode, a different position of the test gas feed inlet (here, the test gases were not fed through the humidifier), and no gas analysis system. In the present study, a differential cell (Baltic qCF type with automotive linear-channel flow field) with an active area of 20.25 cm^2^ was employed, which allowed for the minimization of in-plane effects such as gradients in partial gas pressures, relative humidity, and temperature and therefore enhanced focus on the contamination effect at a given concentration. Although the effects of very low concentrations of impurities eventually become less visible in such a cell [17], a rather uniform coverage of the contaminant on the catalyst throughout the active area was expected.

Low concentrations of impurities were achieved by mixing precontaminated test gases with neat H_2_. Therefore, carbon monoxide (CO, 10 ppm in H_2_ 5.0) and hydrogen sulfide (H_2_S, 0.5 ppm in N_2_ 5.0) were mixed via mass flow controllers with house-supply high-purity hydrogen (all gases provided by Linde AG) in the required fractions.

### 2.2. Materials

The variations in the anode-loading on the catalyst-coated membranes (CCMs, provided by Greenerity GmbH) were achieved through different thicknesses of the anode catalyst layers (CLs), while the cathode loading was kept constant at 400 µg/cm^2^. The catalyst material for both electrodes, the anode and the cathode, was pure Pt on carbon. The membrane electrode assembly (MEA) specifications are shown in Table 1.

### 2.3. Testing Procedure and Conditions

For every test with a different type of contaminant gas, a fresh MEA sample was assembled into the test cell. To measure the effect of the impurities, the test cell was operated with a constant load to detect the voltage drop associated with the contaminant species and concentration. In the following figures, the cell voltage drop is defined as the relative change based on the initial cell voltage. The effect of CO was tested at three different concentrations, namely 0.1, 0.2, and 0.4 ppm (50%, 100%, and 200% of the impurity limit noted in the H_2_ Quality Standard). Before and after the actual contamination, the fuel cell was operated with neat H_2_ to establish a baseline voltage and to detect eventual irreversible degradation of the electrodes. The effect of H_2_S was tested at two concentrations, which were 4 and 20 ppb (100% and 500% of the limit in the Quality Standard), with neat H_2_ operation only at the start of the contaminant test. The conditions during the contaminant tests are shown in Table 2.

The MEAs were characterized, including cyclic voltammetry (CV) on the anode and cathode side at the beginning and end of life (BOL and EOL), as were the polarization curves at the BOL, to compare the performance between the MEA types before starting the contaminant test. The gas pressure during contamination was selected in reference to the studies by Hashimasa et al. [12,14], while the pressure during the polarization curves was chosen according to in-house standardized testing protocols. The conditions during the polarization curves are shown in Table 3.

CV measurements were performed to determine the electrochemically active surface area (ECSA) of the CLs before and after contamination and recovery procedures, specifically from H_2_S poisoning. The CVs were performed on both the anode and cathode electrodes under the conditions summarized in Table 4. To conduct an anode CV, the test cell was purged with nitrogen in order to exchange the gas supply and the electric connectors of the anode and cathode compartment and then reconditioned with fully humidified H_2_ and N_2_ for 12 min prior to the CV. Following the anode CV, the cell was purged again and reconnected in a regular anode/cathode configuration for subsequent tests.

An upper CV boundary of 700 mV was selected to avoid the oxidation of adsorbed foreign species, especially during the H_2_S recovery tests, and to solely focus on the recovery from SD/SU processes. Moreover, the N_2_ flow was stopped during the actual CV to avoid disproportionally high H_2_ evolution currents during the anodic sweep, which were observed especially for the lowest anodic loading. Figure 2 shows exemplary BOL CVs of the three different anode electrodes and one cathode electrode for comparison.

Normally, the ECSA is determined through integration of the charge transfer between voltage boundaries, starting from ~0.08 to 0.1 V to the minima or maxima of the respective double-layer charging current, which typically is somewhere between 0.3 and 0.6 V [18]. However, in this study, these boundaries were considered less suitable for CVs on ultralow-loaded anode CLs. High currents associated with H_2_ evolution during the cathodic sweep (*H_2,ev_*) and the coherent reverse-transport of eventually evolved H_2_ during the anodic sweep (*H_2,rtr_*) would account for relatively large errors in the ECSA. Hence, the voltage boundaries for the determination of the ECSA were chosen as 0.15 to 0.3 V, as shown in Figure 3.

Using this narrowed voltage range, the anode ECSA was determined from the anodic sweeps associated with the adsorption of H_2_ on the catalyst surfaces. Although this procedure cuts the measured ECSA compared to integration between regular voltage ranges, it was found that it would increase the accuracy of the ECSA determination and its changes in the case of ultralow-loaded anodes (as tested in the present study).

## 3. Results and Discussion

### 3.1. Performance and Stability of Ultralow-Loaded Anodic CLs

Before the actual contamination tests, the BOL performance and voltage stability of the MEA samples with ultralow-loaded anodes were established. Figure 4 shows the BOL polarization curves of the three different MEAs when neat H_2_ was supplied to the test cells.

As can be seen in the figure, the polarization curves of the different MEAs overlap quite well, indicating that overpotentials arising due to a lack of active catalyst sites for the hydrogen oxidation reaction (HOR) were not significant for ultralow anodic loadings. In fact, MEA type C (15 µg/cm^2^) even showed a slightly better performance at current densities above 2.5 A/cm^2^, (~15 mV at 3 A/cm^2^), which might have been a result of minimal differences in humidification characteristics of this specific sample and the lower measured high-frequency resistance (HFR).

In addition to the BOL performance, the cell voltage stability of the three MEA types over a testing time of 100 h of continuous operation at a constant load with neat H_2_ was established, which is shown in Figure 5.

During these stability tests, no significant difference between the voltage drops of the MEA types was observed. A slight voltage drop during the first ~2 h was visible for all three MEA types and was associated with the consumption of reactants, which saturated in the electrode before the current was increased.

Overall, the comparability of the different MEA types at the BOL under operation with neat H_2_ was considered satisfactory and was accepted for subsequent tests with contaminants. Before each contamination test, the cell was operated for 20 h with neat H_2_ to establish a baseline voltage. In the case of CO, the first concentration of contaminant was introduced and increased at time steps of 20 h, before we finally shut off the impurity for an additional 20 h of operation with neat H_2_. In the case of H_2_S, after operation with neat H_2_, a single concentration of H_2_S was introduced until the cell voltage broke down, and subsequently SD/SU recovery tests were conducted. For all tests, the anode bubbler required a refill with fresh deionized (DI) water every 10 h. This DI water contained dissolved O_2_, which was driven out as soon as it was heated in the bubbler and was consequently available for the recovery of poisoned Pt sites, which is visible as voltage peaks in the following figures.

### 3.2. Effect of CO on Ultralow-Loaded Anode CLs

Essentially, CO adsorbs on Pt and thereby competes with the actual HOR for active sites on the catalyst surfaces, as shown in Equations (1)–(3):(1)$$2Pt+H_{2}\overset{}{↔}2\left(Pt-H\right),$$
(2)$$Pt+CO\overset{}{↔}Pt-CO,$$
(3)$$2Pt+CO\overset{}{↔}{\left(Pt\right)}_{2}=CO.$$

Depending on the coverage of CO, each molecule blocks one or two active Pt sites via linear or bridge bonds (Equations (2) and (3), respectively) [5,19]. At lower coverages, a higher fraction of bridge bonds is expected, while at higher coverages, an adlayer with CO linear bonds dominates [20]. However, the adlayer CO structure depends on particle sizes, adsorption potentials, facet orientations, and temperature in a complex way because dipole–dipole interactions are important [21]. The effect of different CO concentrations on the voltage decay rates of the three ultralow-loaded anodic CLs is shown in Figure 6.

As expected, the effect of CO in the fuel generally increased for lower anodic loadings, including both a faster and more severe voltage drop. The leveling of the potentials, i.e., the initial decline toward a plateau, depended on the contaminant concentration and the CL thickness [22,23]. For thinner CLs, the reaction front increasingly corresponded with the actual CL thickness, and therefore the local potential was more uniform while contaminants competed throughout the layer with hydrogen for adsorption sites, which resulted in a lower tolerance for thinner (and lower-loaded) CLs. At the ISO concentration (0.2 ppm), the voltage loss due to CO poisoning accounted for ~8%, 41%, and 51% when the anodic loading decreased from 50 to 25 and 15 µg/cm^2^, respectively. Slight potential oscillations of the ultralow-loaded anode MEA types (type B and especially C) at high CO concentrations between normalized voltage ratios of 0.4 and 0.6 were also visible. At these potentials, overpotentials induced by CO poisoning forced the anode potential to shift frequently toward the cathode potential and close to the oxidation potential of CO to CO_2_, allowing for recovery of the electrode [24,25]. This self-recovery was the reason for the maximum coverage of the catalyst with CO in regular PEMFC operation and a flattening of the relative potential drop for lower anodic catalyst loadings with higher CO concentrations, which is partially visible in Figure 7.

In the figure, relative voltage drops due to CO poisoning over the anode Pt loading from the study by Hashimasa et al. and the present study are compared. Although the test cells and the operational parameters between the two studies were different (70% fuel usage in the single cell by the Japanese Automobile Research Institute, JARI, versus 8.3% fuel usage in the differential single cell employed in the present study), a general trend for voltage decay with lower anodic loadings or higher CO concentrations can be seen. The onset of the mentioned flattening of the relative voltage drop at maximum CO coverage is visible for the lowest anodic loading and the highest tested CO concentration, where the relative change between MEA types B and C was less significant compared to types A and B.

In general, CO contamination is fairly easy to mitigate by providing O_2_ to the anode via the air-bleeding technique [3]. This technique not only mitigates CO poisoning, but also partially mitigates poisoning from other contaminants, such as H_2_S [16]. However, as discussed above, air bleeding also comes with disadvantages, such as a reduction in fuel efficiency and potential effects on the integrity of the ionomer in the CLs and membrane. Therefore, to minimize potentially amplified side effects from such mitigation strategies, a reduction of the limit for CO in the H_2_ Quality Standard could be a reasonable option considering the severity of CO poisoning on ultralow anodic loadings, as they likely will be employed in the near future in automotive PEMFCs.

### 3.3. Effect of H_2_S on Ultralow-Loaded Anode CLs

In contrast to CO, H_2_S poisons catalyst surfaces irreversibly through dissociative adsorption on Pt via chemical or electrochemical reaction pathways, as indicated by Equations (4) and (5), respectively. The elemental sulfur on Pt cumulatively occupies active catalyst sites also via linear or bridge bonds, which eventually leads to a complete breakdown of the PEMFC performance [6,14,16]:(4)$$Pt+H_{2}S\overset{}{↔}Pt-S+H_{2},$$
(5)$$Pt+H_{2}S\overset{}{↔}Pt-S+2H^{+}+2e^{-}.$$

Higher catalyst loadings provide a higher nominal ECSA and therefore a larger buffer versus such a breakdown. This decrease in tolerance with a reduction in platinum loading is partially visible in Figure 8, which shows the operation times until the breakdown was observed for ultralow-loaded anodes.

The voltage breakdowns for the highest tested anodic loading (50 µg/cm^2^) were not fully observed. In the case of 4 ppb of H_2_S, the test was purposely interrupted after 340 h of contaminant introduction, as a voltage breakdown was not expected anymore. However, subsequent CVs revealed an almost completely sulfur-blocked ECSA, which is shown in the following sections. In the case of 20 ppb of H_2_S, the test station automatically stopped at the onset of the breakdown after about ~70 h, but the start of the breakdown was still visible.

Interestingly, for both MEA types with ultralow anodic loadings (MEA types B and C), voltage breakdowns were detected after almost similar poisoning times for both tested H_2_S concentrations of 4 and 20 ppb. In Figure 9, which compares the accumulated H_2_S supplied until a 30-mV voltage loss was detected in the present study versus the study by Hashimasa et al., these similar poisoning times are visible as a nonproportional decline in the amount of H_2_S supplied with the reduction in anodic loading.

Although Hashimasa et al. described their observed decline as proportional to the reduction in the loading, their data actually rather showed a slight flattening of the curve with the decrease in the anodic loading, comparable to the data from the presented study. Again, although the test cells and the operational parameters were different (70% fuel usage in JARI’s single cell versus 8.3% fuel usage in the differential single cell in the present study), the general trend was still visible.

One explanation could be that some of the H_2_S adsorbed on the surfaces of the test bench and cell components before actually reaching the CCM and catalyst sites. Depending on the chronology of the tests, this latency could create delays in the voltage breakdown. On the other hand, in the present study, the CVs of lower-loaded anodes also revealed a higher degree of self-recovery from simple shut-down (SD) and start-up (SU) processes.

For these self-recovery tests, the ECSA of the anode CLs exposed to H_2_S were determined at the BOL after a simulated SD/SU process, after H_2_S poisoning, and again after an SD/SU process. The SD/SU included a short purge with dry nitrogen to avoid open circuit voltage (OCV) in H_2_/air-atmosphere, a cooldown of the cell to 20 °C, a wait time of 3 h, and finally again heating of the cell to 80 °C and the introduction of neat H_2_/air to the cell, which was kept at a fixed potential of 0.8 V during the heating. Figure 10 presents these anode CVs for the three different anodic loadings.

Clearly visible is the difference between the CVs at the BOL and after H_2_S contamination (black to yellow CV) for all three MEA types, indicating the reduction of ECSA due to sulfur adsorbed on Pt. For MEA type A, the CV after SD/SU and before H_2_S contamination (blue CV) is additionally shown to exemplarily demonstrate that the SD/SU process did not significantly affect the CV measurement and ECSA determination, as both CVs overlapped quite well. However, when the SD/SU process was carried out after H_2_S contamination, the CV and therefore the ECSA gained in area compared to the poisoned ECSA (yellow to green CV), indicating a partial recovery from previously deactivated ECSA. This self-recovery was increasingly observed with the reduction in the anodic loading. Table 5 presents the nominal ECSAs and percentage changes between the test SD/SU steps.

While only about 35% of the ECSA from MEA type A (50 µg/cm^2^) could be recovered, 84% and almost a full recovery of 94% could be achieved for MEA types B (25 µg/cm^2^) and C (15 µg/cm^2^), respectively, through a simple SD/SU process.

The reason for the different behavior of ultralow-loaded anodes with respect to their tolerance versus H_2_S contamination and the improved self-recovery during SD/SU processes might have a dimensional character in combination with the scavenging effect of water versus contaminants [26]. Studies in the literature investigating the recovery of sulfur-poisoned electrodes have often employed CV-like processes to increase the potential and thereby oxidize adsorbed sulfur either on cathode or anode electrodes [27,28]. During this oxidation, sulfur oxides such as sulfur dioxide (and in combination with water-soluble anions such as sulfate (SO^2-^_4_) or sulfite (SO^2-^_3_)) develop as shown in Equations (6)–(8) [16]:(6)$$Pt-S+O_{2}\overset{}{↔}Pt+SO_{2},$$
(7)$$Pt-S+3H_{2}O\overset{}{↔}Pt+SO_{3}^{2-}+6H^{+}+4e^{-},$$
(8)$$Pt-S+4H_{2}O\overset{}{↔}Pt+SO_{4}^{2-}+8H^{+}+6e^{-}.$$

Presumably, during an SD/SU process, the catalyst surfaces and adsorbed species relax, the local potential varies depending on the local equilibrium and the available species on Pt, and chemical reactions occur to the point of the formation of sulfur anions in the presence of water. It should be noted that the potential of the anodic electrode prior to and during the SD can affect the reduction state of the sulfur species, which eventually facilitates their oxidation or desorption [29]. As the different anodic loadings tested in this study were achieved through variations in CL thickness, the anode of MEA type C consequently had the lowest thickness, while the active cell area remained the same for all samples. During an SD, water condensates and eventually is driven out through hydrophobic pores of the microporous and gas diffusion layer (MPL/GDL) or collects in pores and areas, which are energetically favorable. As the interface between the MPL and CLs also contains such pores [30], sulfur in proximity to this interface might dissolve in these water accumulations in the form of soluble sulfur anions [26]. As the active cell area and therefore the CL/MPL interface area should be the same on average for all three MEA types, while the anode CL volumes are different, a higher fraction of anions could get removed for lower-loaded and therefore thinner anode electrodes. These anions dissolved in water eventually are flushed out once the PEMFC is started again. This works better so long as sulfur is weakly bonded to the Pt surface via linear bonds. With time, adsorbed sulfur develops stronger bonds to active sites and is bound more strongly to the catalyst, leading finally to the observed voltage breakdowns of the PEMFCs. Thinner CLs may also be associated with a changed ionomer structure, and the potentials within the layer are generally more homogeneous [31]. However, the differentiation of this effect is beyond the scope of this paper.

Consequently, the reduction of the anodic catalyst material down to ultralow loadings seemed to come with a nonproportional reduction in tolerance versus H_2_S poisoning and an improved self-recovery during SD/SU processes. Hence, lowering the ISO limit for sulfur-containing compounds might not be necessary with regard to ultralow-loaded anode electrodes. However, these effects should be further confirmed in large- or full-scale cell tests using realistic automotive fuel utilizations.

## 4. Conclusions

The key findings from this study are that the H_2_ Quality Standard ISO 14687-2:2012 eventually requires partial adaption to accommodate future automotive PEMFC designs, including ultralow-loaded anodic CLs, and that ultralow-loaded anodes exhibited an improved self-recovery from sulfur poisoning from simple SD/SU processes.

As expected, CO poisoning induced significant performance losses at an increasing rate and severity with decreases in the platinum loading. At an ISO concentration of 0.2 ppm CO in the fuel, the cell voltage was about 40–50% lower compared to operation with neat H_2_ for ultralow anodic loadings, which raises the question of whether the CO limit in the H_2_ Quality Standard needs to be reduced with regard to future anodic loadings.

When H_2_S was in the fuel, the ultralow-loaded anodic CLs exhibited a nonlinear reduction as opposed to the expected linear reduction in tolerance to the reduction in platinum loading. Simultaneously, these anodic CLs recovered to larger degrees from sulfur poisoning during the SD/SU processes. It is hypothesized that the nonlinear reduction in tolerance and improved self-recovery arose due to the decrease in the ratio between the CL thickness (and coherent ECSA) and the geometric cell area. As the ultralow-loaded anodes were also the thinner CLs, larger fractions of sulfur adsorbed on catalyst surfaces in proximity to pores at the CL–MPL interface could have dissolved in the water present in the form of anions, which were driven out of the cell during operation or during the SU of the PEMFCs.

However, to confirm these findings, the performance of ultralow-loaded anodic CLs in the presence of impurities should be further investigated, ideally in large- or full-scale PEMFCs using automotive fuel consumption rates.

## Figures and Tables

**Figure 1 molecules-24-03514-f001:**
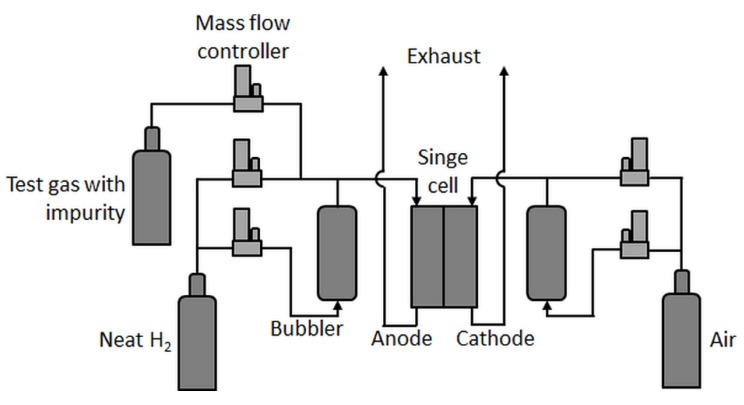
Single-cell test station scheme.

**Figure 2 molecules-24-03514-f002:**
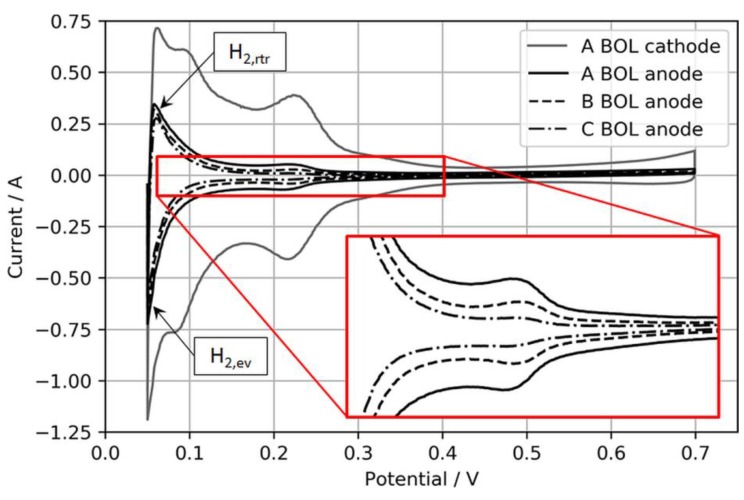
Anode CVs of MEA types A, B, and C with an MEA type A cathode CV for reference. The inset expands the H_2_ adsorption/desorption regions of the anode catalyst layers (CLs) for visual comparison.

**Figure 3 molecules-24-03514-f003:**
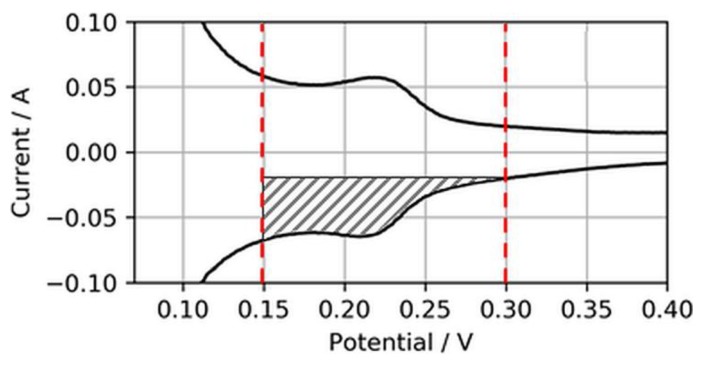
Electrochemically active surface area (ECSA) determination from reduced H_2_ adsorption area.

**Figure 4 molecules-24-03514-f004:**
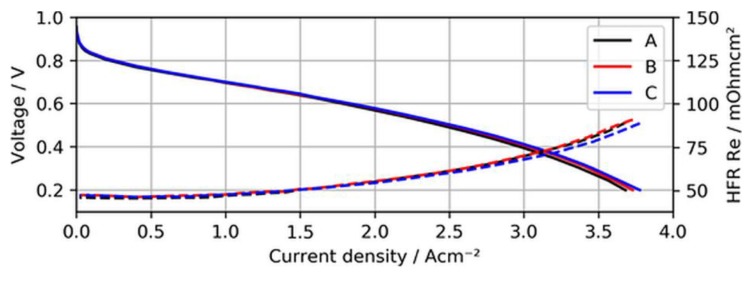
Polarization curves of MEA types A, B, and C using neat H_2_, with high-frequency resistance (HFR) as dashed lines.

**Figure 5 molecules-24-03514-f005:**
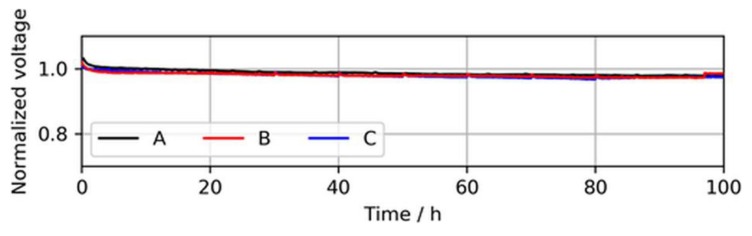
Voltage decay over 100 h of continuous operation in neat H_2_ at 1.0 A/cm^2^, showing similar voltage stabilities of the MEA types. The voltages were normalized to the initial cell voltage at time = 0 h.

**Figure 6 molecules-24-03514-f006:**
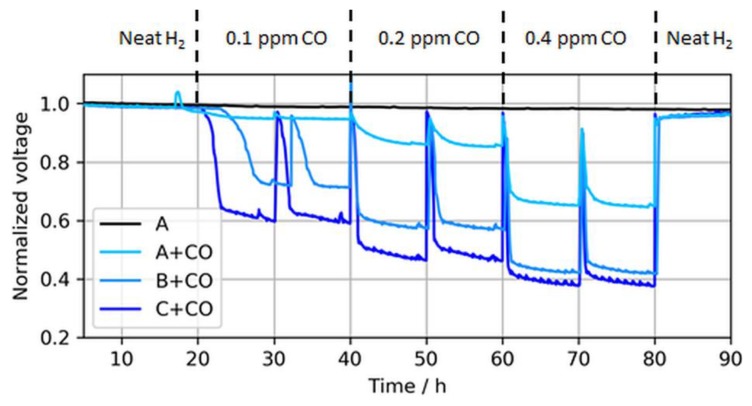
Voltage drops induced by different CO concentrations in MEA types A, B, and C at a constant load of 1.0 A/cm^2^. The voltage peaks (every 10 h) were caused by anode bubbler refills and coherent recovery of Pt sites with O_2_ dissolved in DI water. Again, the voltages were normalized to the initial cell voltage at time = 0 h, while the results are shown starting from *t* = 5 h.

**Figure 7 molecules-24-03514-f007:**
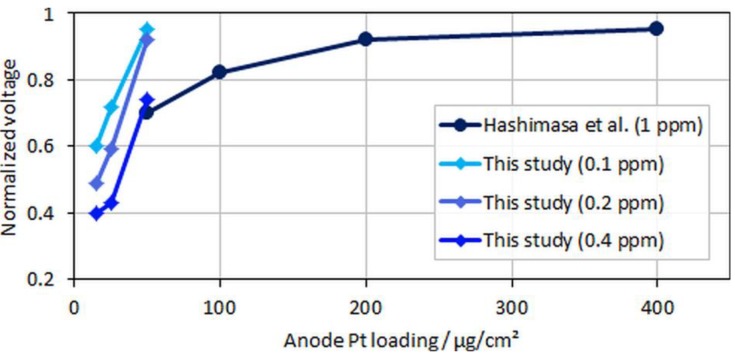
Normalized voltages over anodic loading; data adapted from Hashimasa et al. [12].

**Figure 8 molecules-24-03514-f008:**
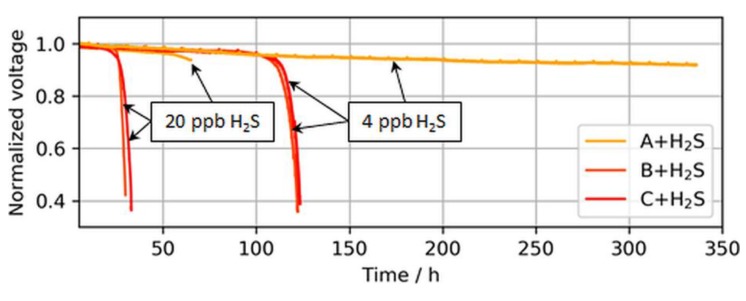
Voltage breakdowns induced by 4 and 20 ppb of H_2_S during operation at a constant load of 1.0 A/cm^2^. The operation of MEA type A was purposely stopped after ~340 h and ~70 h, while MEA types B and C stopped automatically after voltage breakdowns were observed.

**Figure 9 molecules-24-03514-f009:**
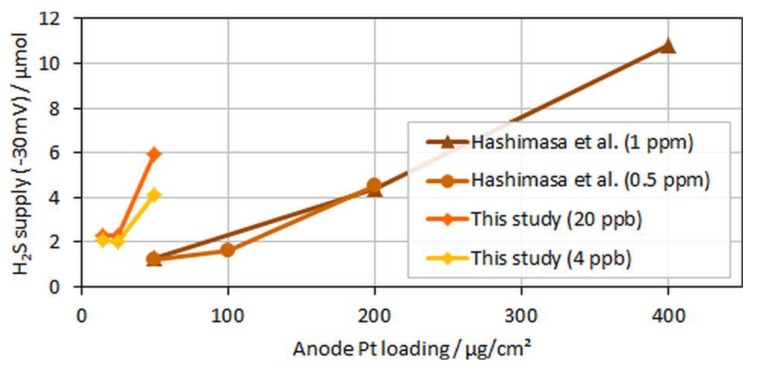
H_2_S supplied to the cell until voltage dropped by 30 mV over anodic loading; data adapted from Hashimasa et al. [14].

**Figure 10 molecules-24-03514-f010:**
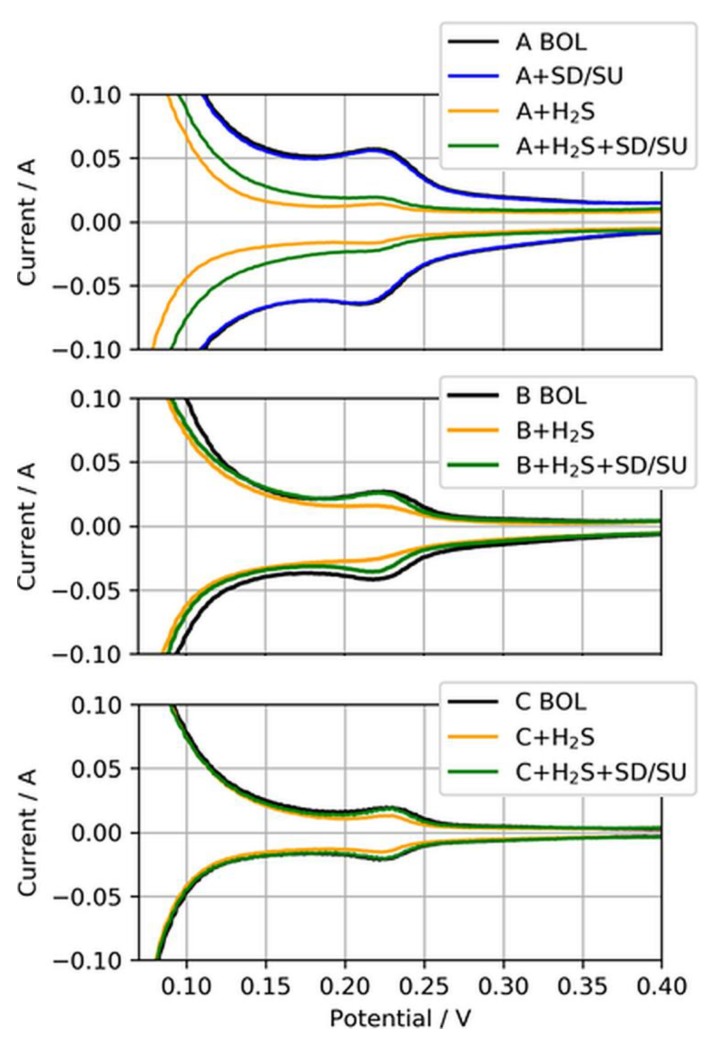
Anode CVs for MEA types A, B, and C (50, 25, and 15 µg/cm^2^) at the beginning of life (BOL), after H_2_S contamination (+H_2_S), and after a subsequent shut-down/start-up (SD/SU) process (+H_2_S + SD/SU). For MEA type A, the CV after SD/SU just before the contamination is also shown (A + SD/SU).

**Table 1 molecules-24-03514-t001:** Membrane electrode assembly (MEA) specifications.

Active Cell Area		20.25 cm^2^
Catalyst	Anode	Pt/C
	Cathode	Pt/C
Electrode loading	Anode	50/25/15 µg/cm^2^ (named A, B, and C hereafter)
	Cathode	400 µg/cm^2^
Membrane thickness		~15 µm
Gas diffusion layer		Freudenberg H23C9

**Table 2 molecules-24-03514-t002:** Operating conditions during contamination.

Cell Temperature	80 °C
Outlet pressure anode/cathode	1.2/1.2 bara
Relative humidity anode/cathode	90%/75%
Current density	1.0 A/cm^2^
Stoichiometry anode/cathode	12/14

**Table 3 molecules-24-03514-t003:** Polarization curve conditions.

Cell Temperature	80 °C
Outlet pressure anode/cathode	2/2 bara
Relative humidity anode/cathode	95%/75%
Gas flow anode/cathode	3/7 l/min

**Table 4 molecules-24-03514-t004:** Cyclic voltammetry (CV) conditions.

Cell Temperature	80 °C
Outlet pressure anode/cathode	1.05/1.05 bara
Relative humidity anode/cathode	95%/95%
Gas flow anode/cathode	1.0/0 l/min (1.0 l/min N_2_ for 12 min prior to CV on cathode)
Scan range	50–700 mV
Sweep rate	100 mV/s

**Table 5 molecules-24-03514-t005:** ECSA at the BOL and relative change after shut-down/start-up processes (SD/SU) before and after contamination with H_2_S based on narrowed boundaries (integration between 150 and 300 mV). Note: the nominal ECSA was lower by about 60–70% than what would be typically expected for the specific catalyst material, while the relative ECSA changes were amplified to some degree due to the narrowed voltage boundaries and therefore the smaller area for integration.

MEA Type	ECSA (m^2^/g Pt)			
	BOL	After SD/SU	After H_2_S	After H_2_S + SD/SU
A	20.5	20.0 (98%)	4.4 (22%)	7.1 (35%)
B	24.4	24.2 (99%)	16.8 (69%)	20.6 (84%)
C	19.6	19.1 (97%)	14.0 (71%)	18.4 (94%)

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
