# Peer review of "Tolerance and Recovery of Ultralow-Loaded Platinum Anode Electrodes upon Carbon Monoxide and Hydrogen Sulfide Exposure"

_molecules, 2019, doi:10.3390/molecules24193514_

Round 1

Reviewer 1 Report

This paper describes the first study investigates the tolerance of  the anodic Pt catalyst for fuel cells with an ultra-low anodic loadings against CO and H2S. The air bleeding technique was also applied to understand the effectiveness of the method.

The paper is important for the application of the PEFCs in the near future. The experiments have been very carefully carried out, and the discussions and conclusions are fairly conducted.

This paper should be published.

Minor comments:

1) Please briefly describe the gas channel type of Baltic qCF-type cell.

2) Page 4 Line 105  A parenthesis needed as "Therefore, carbon monoxide (CO, 10 ppm in H2 5.0) and".

Author Response

Dear Reviewer,

Thank you very much for the revision and rating of the manuscript. We incorporated the comments as follows:

The word “automotive” was added to the Baltic qCF-type cell description. Unfortunately, more details on the flow field design cannot be given, as the test cell is based on former AFCC design and Daimler proprietary information. Although it is always good to know the flow field design of such cells, we think that the detailed design should not be the determining factor in this contamination study using a differential test cell. The missing parenthesis was added.

Kind regards,                                     

Sebastian Prass

Reviewer 2 Report

This paper study the effects of CO and H2S on the performance of PEMFC. Experimental techniques were used to conduct this investigation. Overall, the paper is written well and the reported results are good. However, many issues still exist in the current paper and need to be addressed before recommending the paper for any possible publication.
- It is recommended to add subsections to section 2 (Material and methods) as this section is currently very long
- Please correct the number of section 3.2 (It should be 3.1) in page 5
- In conclusion section, the innovative aspect of this work should be clearly underlined with respect to the other similar works in the literature.
- Please add a a recommendation to put the conclusions in the frame of a future work.

Author Response

Dear Reviewer,

Thank you very much for the revision and rating of the manuscript. We incorporated the comments as follows:

The subsections were added as suggested. The numbers were corrected. The difference to the most relevant study in literature with respect to sulfur tolerance of low and ultra-low loaded anodes (non-proportional instead of proportional decline with reduction in platinum loading) was  stressed in section 3.3 and the conclusion section. The last sentence in the conclusion section was modified to emphasize the recommendations for future works, which could involve contamination tests in large or full scale fuel cells using automotive fuel consumption rates.

Kind regards,                                     

Sebastian Prass